# Conversational Few-Shot Prompting: Rethinking Few-Shot Prompting for Chat Language Model

## Abstract

In-context learning, also referred to as few-shot learning, enables language models to adapt to tasks using a limited number of examples embedded in the prompt. Traditional approaches typically present all examples in a single prompt, which works well for pre-trained base models. However, the application of this method to instruction-tuned chat models, such as ChatGPT, remains underexplored. In this paper, we introduce a novel *conversational few-shot prompting* technique, which structures few-shot examples as multi-turn conversation between the user and the assistant, rather than a single input prompt. This conversational framing better aligns with the interactive nature of chat models, enhancing their instruction-following abilities and generalization across tasks. Through experiments on various benchmarks, we demonstrate that this approach significantly improves performance, particularly in low-shot scenarios, compared to traditional few-shot prompting. Our results suggest that this method provides a more flexible and robust way to leverage few-shot examples in instruction-tuned chat models, improving task performance without the need for additional fine-tuning, reducing prompt sensitivity, and offering potential for diverse applications.

## 1 Introduction

In-context learning, particularly in the realm of few-shot learning (Brown et al., 2020), marks a notable progression in language modeling (Kaplan et al., 2020). This method enables models to adjust to specific tasks using only a small number of examples included directly in the input, contrasting with traditional methods that necessitate explicit fine-tuning for each task (Devlin et al., 2019; Raffel et al., 2020). This technique is prominently applied by advanced large language models such as GPT (OpenAI, 2022; Achiam et al., 2023) and Claude (Anthropic, 2023a;b), which dynamically generalize and execute tasks based on limited example encoded in the prompt. Utilizing these examples, the model identifies task patterns and extends them to novel, unseen data in the same context. This approach provides a flexible and computationally efficient alternative to conventional training methods, enabling a single model to handle diverse tasks without extensive labeled data or separate fine-tuning stages.

In the structured arrangement of few-shot learning examples, each instance comprises input-output pairs as depicted in Figure 1. Each example initiates with a constant symbol token, such as "Question" or "Answer" (Brown et al., 2020) to categorize the subsequent content's function. The collection of few-shot examples, along with the user query, is concurrently processed by the language model, which then generates predictions that emulate the provided examples. Employing these fixed symbol tokens to illustrate the concepts of context and completion is effective for a pre-trained base language model, which mainly referring to model without supervised instruction tuning (SFT) (Wei et al., 2022a) or reinforcement learning from human feedback (RLHF) (Ouyang et al., 2022; Bai et al., 2022). However, this approach may not be optimal for an instruction-based chat model conducting SFT or RLHF. For chat models, we apply a chat template that employs specialized tokens "`<|user|>`" and "`<|assistant|>`" to differentiate between context and completion (OpenAI, 2022) as shown in Figure 1. All the input information would be fed into the content of user message "`<|user|>`" and chat model will fill the generation after the "`<|assistant|>`". In practical applications of few-shot prompting, all few-shot examples are incorporated into the user's message

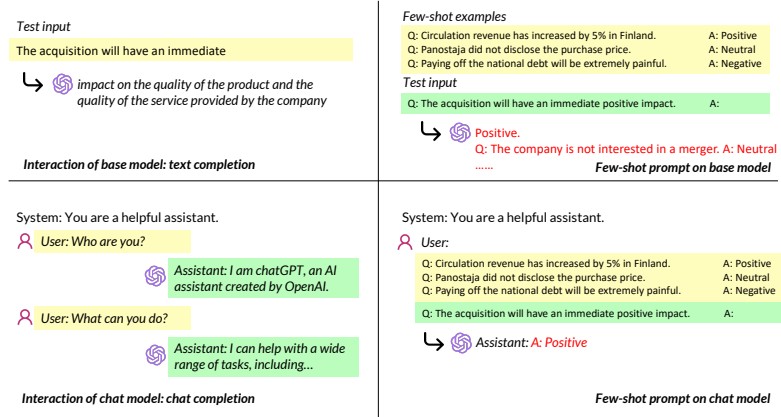

Figure 1: Distinct paradigms of few-shot prompting applied to the base model and chat model.

such as Figure 1, facilitating only a single-turn interaction between the user and the assistant model. Obviously, the task of generation becomes a single-response completion task rather than an ongoing conversation, which may not fully capitalize on the potential of chat models.

In this paper, we investigate a new conversational few-shot prompting technique which adjusts the few-shot prompt as a multi-turn conversation between user and assistant. Rather than incorporating all few-shot examples in a single user message, we convert each input-output pair into a separate turn within the conversation between the user and the assistant as shown in Figure 2. This approach offers several advantages over traditional few-shot learning methods. Firstly, by structuring the examples as a dialogue, has the potential to improve the performance and usability of few-shot learning in chat models, making them more versatile and efficient in handling a wide range of tasks with limited examples. Secondly, this technique aligns more naturally with the interactive nature of chat models, allowing them to leverage their inherent conversational capabilities to better follow user's instruction. Finally, the approach is task-agnostic, and it be used across a variety of tasks and models without requiring task-specific fine-tuning provided the model has been trained using supervised instruction tuning.

To empirically validate these potential benefits, we propose a series of experiments comparing the performance of traditional few-shot prompting with our conversational approach across various tasks and models. These experiments will measure factors such as task accuracy, instruction following ability, and chain of thought ability. The objective of this research is to augment the extant literature on few-shot learning in language models by investigating this conversational method. This approach holds the potential to substantially influence the advancement and utility of chat models across diverse domains including customer service, educational support, and other sectors.

## 2 PRELIMINARIES

### 2.1 FEW-SHOT PROMPTING ON PRE-TRAINED BASE MODEL

Scaling up the size of language models has been shown to confer a range of benefits, such as improved performance and advanced reasoning ability (Kaplan et al., 2020). Large language models offer the exciting prospect of in-context few-shot learning via prompting (Brown et al., 2020). That is, instead of finetuning a separate language model checkpoint for each new task, one can simply "prompt" the model with a few input–output exemplars demonstrating the task. The traditional few-shot prompting method used in recent works is designed for pretrained base model, as demonstrated below.

#### 2.1.1 DEFINITION

For pre-trained base language model $\mathcal{M}_{base}$:

- $X$ represent the input space, which is the set of all possible inputs.
- $Y$ represent the output space, which is the set of all possible outputs.

The pre-trained language model $\mathcal{M}_{base}$ generates the output $y \in Y$ based on the input $x \in X$:

$$y = \mathcal{M}_{base}(x). \tag{1}$$

### 2.1.2 APPLICATION ON FEW-SHOT PROMPTING

Given a task $T$, a set of $k$ examples is provided in the form of input-output pairs: $\{(x_1, y_1), (x_2, y_2), \ldots, (x_k, y_k)\} \subset X \times Y$. These examples are then combined into a prompt $p \in \mathcal{P}$, where $\mathcal{P}$ is the set of all possible prompts.

The structure of the prompt is defined as:

$$p = [(x_1, y_1), (x_2, y_2), \ldots, (x_k, y_k)]. \tag{2}$$

The pre-trained language model $\mathcal{M}$ generates the output $y_{\text{new}} \in Y$ corresponding to the new input $x_{\text{new}}$ based on the provided prompt:

$$y_{\text{new}} = \mathcal{M}_{base}(x_{new}, p). \tag{3}$$

## 2.2 FEW-SHOT PROMPTING ON CHAT MODEL

In a chat-based model, the conversation is modeled as a sequence of message exchanges between different roles (OpenAI, 2022; Touvron et al., 2023; Dubey et al., 2024). Each message is represented as a tuple containing the role of the participant and the content of the message. The model generates responses based on the conversation history.

### 2.2.1 DEFINITION

For chat-based model $\mathcal{M}_{chat}$:

- $U$ represent the set of all possible user inputs.
- $A$ represent the set of all possible assistant responses.
- $S$ represent the set of all possible system messages.
- $\mathcal{R}$ represents the role of the speaker, where $\mathcal{R} = \{\texttt{system}, \texttt{user}, \texttt{assistant}\}$.

The conversation can be viewed as a sequence of interactions between the user and the assistant. Conversation context $\mathcal{C}_t$ at any given turn $t$ is represented as:

$$\mathcal{C}_t = [(r_0, s), (r_1, u_1), (r_2, a_1), \ldots, (r_1, u_t), (r_2, a_t)], \tag{4}$$

where $u_t \in U$ is the user input at turn $t$, $a_t \in A$ is the model response at turn $t$ and $s \in S$ represent a possible system message. Specifically, $r_i \in \mathcal{R}$ represents different role of the speaker, where $r_0 = \texttt{system}$, $r_1 = \texttt{user}$ and $r_2 = \texttt{assistant}$.

The assistant's response $a_t$ from a chat-based model $\mathcal{M}_{chat}$ is generated based on both the current user input $u_t$ and the conversation context $\mathcal{C}_{t-1}$:

$$a_t = \mathcal{M}_{chat}(r_1, u_t, \mathcal{C}_{t-1}). \tag{5}$$

### 2.2.2 APPLICATION OF FEW-SHOT PROMPTING

For chat-based models, few-shot prompts should be included in the chat template as part of the user's message. They can be represented as:

$$p = [(r_0, s), \{r_1, (x_1, y_1), (x_2, y_2), \ldots, (x_k, y_k)\}], \tag{6}$$

where all the few-shot examples $(x_1, y_1), (x_2, y_2), \ldots, (x_k, y_k)$ serve as the message content of the role $r_1 = \texttt{user}$. It is important to note that all the few-shot examples are actually acting only a one-turn interaction between the user and the assistant.

The chat-based model $\mathcal{M}_{chat}$ uses the provided few-shot prompt to generate the new assistant response $y_{\text{new}}$ for the input $x_{\text{new}}$. This is achieved by conditioning the model on the prompt examples:

$$y_{\text{new}} = \mathcal{M}_{chat}(r_1, x_{\text{new}}, p). \tag{7}$$

## 3 CONVERSATIONAL FEW-SHOT PROMPT

The chat template-based few-shot prompting approach combines the strengths of the traditional few-shot prompting and multi-turn conversation structure of chat-based model. This method organizes the examples as a dialogue between a user and an assistant, mimicking the natural flow of a conversation. By presenting the examples in this format, the model can better understand the context and generate more coherent and relevant responses. This approach aligns more effectively with the concept of "few-shot" learning in the setting of chat-based models.

### 3.1 DEFINITION

For a specific task $T$, a collection of $k$-shot examples is supplied. The goal of the multi-turn conversation approach in few-shot prompting is to treat each shot as an interactive turn between the user and the assistant.

*System: You are a helpful assistant.*

*User:* Circulation revenue has increased by 5% in Finland.

*Assistant:* Positive

*User:* Panostaja did not disclose the purchase price

*Assistant:* Neutral

*User:* Paying off the national debt will be extremely painful.

*Assistant:* Negative

*User:* The acquisition will have an immediate positive impact.

*Assistant:* **Positive**

**Conversational few-shot prompt**

Figure 2: Our conversational few-shot prompting approach.

The prompt is then structured as follows:

$$p = [(r_0, s), \{(r_1, x_1), (r_2, y_1)\}, \{(r_1, x_2), (r_2, y_2)\}, \ldots, \{(r_1, x_k), (r_2, y_k)\}]. \tag{8}$$

Given the few-shot prompt $p$, the language model $\mathcal{M}_{chat}$ generates the assistant's response $a_{\text{new}} \in Y$ for the new user message in $u_{\text{new}}$. The model does this by conditioning on the provided few-shot examples as well as the conversation history:

$$a_{\text{new}} = \mathcal{M}_{chat}(r_1, u_{\text{new}}, p). \tag{9}$$

Thus, the assistant's response is generated by considering both the new user input and the patterns demonstrated in the few-shot examples.

## 4 EVALUATION OF PROBABILITY RANKING

### 4.1 EXPERIMENTAL SETUP

**Models** We evaluate powerful open-source models from popular LLM families across various sizes. The first is Llama series (Dubey et al., 2024): Llama-3.2-1B-Instruct, Llama-3.2-3B-Instruct, Llama-3.1-8B-Instruct and Llama-3.1-70B-Instruct. The second is Qwen2 series (Yang et al., 2024), for which we use Qwen2-0.5B-Instruct, Qwen2-7B-Instruct, and Qwen2-72B-Instruct. The third is OLMo series: OLMo-7B-0724-SFT-hf, OLMo-7B-0724-Instruct-hf (Groeneveld et al., 2024). Due to space limits, the results of the Qwen2 series and OLMo series are presented in the Appendix Table 8.

**Benchmark** We conduct experiments on widely used in-context learning benchmarks for classification: SST2, MNLI and BoolQ (Wang, 2018; Wang et al., 2019), and multiple-choice tasks: MMLU (Hendrycks et al., 2021a), MMLU-Pro (Wang et al., 2024). These benchmarks are selected for their diversity in tasks and domains, providing comprehensive evaluation for LLM performance.

**Evaluation** Our evaluation protocol follows the mainstream LLM evaluation frameworks, EleutherAI lm-eval-harness (Gao et al., 2024)[1]. Specifically, for classification and multiple-choices tasks, we access the output probabilities of option tokens and use the maximal one as the model prediction. We utilize an identical few-shot examples paired with instruction templates, adjusting the number of shots from 1 to 5 to ensure the validity of our results.

---

[1]During the writing of this manuscript, the lm-eval-harness introduced support for conversational few-shot prompting, which has been designated as *fewshot_as_multiturn*.

Table 1: Performance of different models in different shot settings (Probability ranking)

| Model | Shots | MMLU-Pro | | MMLU | | SST2 | | MNLI | | Boolq | |
|---|---|---|---|---|---|---|---|---|---|---|---|
| | | fewshot | conv-fewshot | fewshot | conv-fewshot | fewshot | conv-fewshot | fewshot | conv-fewshot | fewshot | conv-fewshot |
| Llama-3.2-1B-Instruct | 5 | 11.4 | 17.3 (+5.9) | 23.5 | 44.3 (+20.8) | 63.9 | 90.8 (+26.9) | 33.7 | 33.4 (-0.3) | 43.8 | 64.5 (+20.7) |
| | 4 | 11.3 | 16.9 (+5.6) | 23.4 | 44.3 (+20.9) | 63.4 | 90.2 (+26.8) | 33.5 | 34.0 (+0.5) | 43.2 | 65.0 (+21.8) |
| | 3 | 11.4 | 17.0 (+5.6) | 23.5 | 44.1 (+20.6) | 63.2 | 88.9 (+25.7) | 33.3 | 33.9 (+0.6) | 46.0 | 65.1 (+19.1) |
| | 2 | 11.4 | 16.7 (+5.3) | 23.4 | 43.7 (+20.3) | 64.3 | 88.3 (+24.0) | 33.3 | 34.0 (+0.7) | 50.1 | 65.3 (+15.2) |
| | 1 | 11.4 | 16.7 (+5.3) | 27.5 | 41.8 (+14.3) | 60.0 | 82.9 (+22.9) | 33.7 | 33.8 (+0.1) | 62.0 | 64.1 (+2.1) |
| Llama-3.2-3B-Instruct | 5 | 11.5 | 27.2 (+15.7) | 29.7 | 61.3 (+31.6) | 84.1 | 91.4 (+7.3) | 39.4 | 52.2 (+12.8) | 71.5 | 81.2 (+9.7) |
| | 4 | 11.7 | 26.4 (+14.7) | 29.6 | 61.7 (+32.1) | 84.6 | 92.1 (+7.5) | 40.0 | 52.0 (+12.0) | 72.0 | 81.9 (+9.9) |
| | 3 | 11.7 | 28.1 (+16.4) | 29.8 | 61.4 (+31.6) | 84.5 | 92.2 (+7.7) | 42.9 | 51.7 (+8.8) | 72.9 | 81.9 (+9.0) |
| | 2 | 11.7 | 28.8 (+17.1) | 30.9 | 61.4 (+30.5) | 81.4 | 91.9 (+10.5) | 42.9 | 50.3 (+7.4) | 74.4 | 82.1 (+7.7) |
| | 1 | 12.2 | 29.0 (+16.8) | 30.1 | 60.7 (+30.6) | 78.3 | 88.8 (+10.5) | 39.4 | 46.5 (+7.1) | 72.6 | 81.4 (+8.8) |
| Llama-3.1-8B-Instruct | 5 | 22.3 | 38.1 (+15.8) | 56.7 | 68.4 (+11.7) | 92.2 | 94.6 (+2.4) | 68.9 | 73.5 (+4.6) | 86.2 | 85.8 (-0.4) |
| | 4 | 22.5 | 37.8 (+15.3) | 55.0 | 68.3 (+13.3) | 91.2 | 94.7 (+3.5) | 68.8 | 73.4 (+4.6) | 86.3 | 86.2 (-0.1) |
| | 3 | 22.5 | 37.9 (+15.4) | 54.9 | 67.8 (+12.9) | 91.2 | 94.4 (+3.2) | 67.4 | 72.1 (+4.7) | 86.5 | 86.5 (+0.0) |
| | 2 | 20.8 | 37.2 (+16.4) | 51.6 | 67.9 (+16.3) | 88.9 | 94.0 (+5.1) | 65.6 | 70.5 (+4.9) | 85.7 | 86.0 (+0.3) |
| | 1 | 15.9 | 35.9 (+20.0) | 51.2 | 67.3 (+16.1) | 88.6 | 92.3 (+3.7) | 59.6 | 65.9 (+6.3) | 83.3 | 84.9 (+1.6) |
| Llama-3.1-70B-Instruct | 5 | 36.9 | 38.8 (+1.9) | 78.1 | 82.1 (+4.0) | 94.2 | 94.5 (+0.3) | 56.6 | 59.7 (+3.1) | 62.3 | 62.2 (-0.1) |
| | 4 | 36.0 | 38.9 (+2.9) | 77.8 | 82.1 (+4.3) | 94.0 | 94.3 (+0.3) | 56.4 | 59.8 (+3.4) | 62.3 | 62.2 (-0.1) |
| | 3 | 35.3 | 39.4 (+4.1) | 76.7 | 82.1 (+5.4) | 93.6 | 94.5 (+0.9) | 54.6 | 59.2 (+4.6) | 62.4 | 62.2 (-0.2) |
| | 2 | 34.2 | 40.9 (+6.7) | 76.5 | 81.9 (+5.4) | 92.5 | 94.4 (+1.9) | 52.2 | 58.0 (+5.8) | 62.2 | 62.2 (+0.0) |
| | 1 | 29.8 | 42.9 (+13.1) | 70.4 | 81.6 (+11.2) | 91.7 | 93.9 (+2.2) | 51.7 | 54.8 (+3.1) | 62.2 | 62.2 (+0.0) |

## 4.2 RESULTS

**Conversational few-shot prompting achieves significant improvement across various tasks and varies with model families and sizes.** This improvement is consistent across various benchmarks, such as MMLU-Pro, MMLU, SST2, MNLI, and Boolq across on classification and multiple-choices tasks. For instance, in the case of the Llama-3.2-1B-Instruct model, conversational few-shot prompting yields an improvement of 20.8 points on MMLU with five shots, compared to the standard few-shot setting. Similar patterns can be observed in SST2, where the performance increases by 26.9 points, and in Boolq, where the improvement reaches 20.7 points. These gains highlight the effectiveness of conversational few-shot prompting in capturing contextual information more efficiently, which is particularly beneficial in complex reasoning tasks.

**Performance improvement is task-dependent.** While the results consistently indicate that conversational prompting leads to an improvement across different tasks, the magnitude of these gains varies. For example, in the MMLU-Pro and MMLU tasks, which involve complex reasoning across multiple subjects, conversational prompting shows large performance jumps for smaller models. In contrast, tasks such as MNLI and SST2, which involve natural language inference and sentiment analysis, show more moderate gains. The smallest improvement is observed in the Boolq task for the Llama-3.2-1B-Instruct model with only 2.1 points in the one-shot setting. This suggests that tasks requiring nuanced understanding or reasoning may benefit more from conversational context, whereas simpler classification tasks may see less drastic improvements.

**Scaling trends in model sizes with conversational prompting.** Another important observation from Table 1 is the trend of scaling in model sizes. As the model size increases, the impact of conversational prompting becomes more stable, and improvements tend to plateau. For example, in the Llama-3.1-70B-Instruct model, improvements on tasks like MMLU and SST2 are relatively small, indicating that larger models may have already internalized much of the knowledge that conversational few-shot prompting helps smaller models to acquire.

**Conversational prompting accelerates model learning with fewer examples.** It reduces the model's reliance on large numbers of examples to achieve optimal performance. In traditional few-shot prompting, models often require a higher number of examples (5-shot or more) to reach their best performance. However, with conversational prompting, even fewer examples (1 or 2 shots) can provide sufficient context for the model to perform well.

**Smaller models benefit less but still show consistent improvements.** While the largest models exhibit the most substantial gains from conversational prompt, smaller models like Llama-3.2-1B-Instruct, Qwen2-0.5B-Instruct shown in Table 8 also benefit, though to a lesser extent. For instance, Qwen2-0.5B-Instruct sees an increase from 15.2% to 15.7% in 1-shot MMLU-Pro, and from 40.9% to 42.4% in MMLU with conversational few-shot. The smaller capacity of models like Qwen2-0.5B-Instruct may limit their ability to fully utilize the additional context provided by the conversational

format, but even these smaller models show consistent performance improvements across benchmarks and shot settings.

Table 2: Performance of different models in different shot settings on MMLU and MMLU-Pro.

| Model | Shots | MMLU-Pro | | | | MMLU | | | |
| | | Strict-Match | | Flexible-Match | | Strict-Match | | Flexible-Match | |
| | | fewshot | conv-fewshot | fewshot | conv-fewshot | fewshot | conv-fewshot | fewshot | conv-fewshot |
|---|---|---|---|---|---|---|---|---|---|
| Llama-3.2-1B-Instruct | 5 | 0.0 | 18.4 (+18.4) | 8.6 | 18.7 (+10.1) | 0.0 | 39.0 (+39.0) | 0.5 | 39.6 (+39.1) |
| | 4 | 0.0 | 18.2 (+18.2) | 8.6 | 18.4 (+9.8) | 0.0 | 38.4 (+38.4) | 0.5 | 39.3 (+38.8) |
| | 3 | 0.0 | 16.3 (+16.3) | 8.7 | 16.5 (+7.8) | 0.0 | 36.3 (+36.3) | 0.2 | 38.0 (+37.8) |
| | 2 | 0.0 | 14.4 (+14.4) | 8.5 | 14.8 (+6.3) | 0.0 | 35.5 (+35.5) | 0.3 | 37.4 (+37.1) |
| | 1 | 0.0 | 11.1 (+11.1) | 4.4 | 11.8 (+7.4) | 0.0 | 29.4 (+29.4) | 0.5 | 34.4 (+33.9) |
| Llama-3.2-3B-Instruct | 5 | 0.0 | 31.3 (+31.3) | 0.1 | 31.5 (+31.4) | 0.0 | 55.6 (+55.6) | 0.1 | 57.0 (+56.9) |
| | 4 | 0.0 | 30.8 (+30.8) | 0.0 | 31.1 (+31.1) | 0.0 | 55.5 (+55.5) | 0.1 | 57.1 (+57.0) |
| | 3 | 0.0 | 31.2 (+31.2) | 0.0 | 31.9 (+31.9) | 0.0 | 54.4 (+54.4) | 0.1 | 56.6 (+56.5) |
| | 2 | 0.0 | 30.0 (+30.0) | 0.0 | 31.2 (+31.2) | 0.0 | 53.1 (+53.1) | 0.2 | 56.9 (+56.7) |
| | 1 | 0.0 | 24.6 (+24.6) | 0.0 | 27.6 (+27.6) | 0.0 | 43.1 (+43.1) | 0.3 | 53.8 (+53.5) |
| Llama-3.1-8B-Instruct | 5 | 0.0 | 34.0 (+34.0) | 5.5 | 38.5 (+33.0) | 0.0 | 67.8 (+67.8) | 7.9 | 68.6 (+60.7) |
| | 4 | 0.0 | 31.1 (+31.1) | 3.8 | 38.1 (+34.3) | 0.0 | 67.2 (+67.2) | 6.3 | 64.8 (+58.5) |
| | 3 | 0.0 | 24.7 (+24.7) | 2.6 | 37.3 (+34.7) | 0.0 | 64.9 (+64.9) | 5.6 | 64.2 (+58.6) |
| | 2 | 0.0 | 12.8 (+12.8) | 1.3 | 34.7 (+33.4) | 0.0 | 56.6 (+56.6) | 3.4 | 64.5 (+61.1) |
| | 1 | 0.0 | 0.8 (+0.8) | 0.4 | 30.9 (+30.5) | 0.0 | 21.9 (+21.9) | 3.3 | 65.4 (+62.1) |
| Llama-3.1-70B-Instruct | 5 | 6.7 | 52.8 (+46.1) | 22.5 | 53.1 (+30.6) | 19.8 | 79.2 (+59.4) | 51.7 | 80.5 (+28.8) |
| | 4 | 5.8 | 52.4 (+46.6) | 19.0 | 52.8 (+33.8) | 16.2 | 78.7 (+62.5) | 47.1 | 80.7 (+33.6) |
| | 3 | 3.4 | 51.1 (+47.7) | 14.0 | 51.8 (+37.8) | 14.5 | 77.0 (+62.5) | 43.4 | 80.3 (+36.9) |
| | 2 | 2.5 | 47.6 (+45.1) | 10.6 | 49.7 (+39.1) | 11.4 | 74.0 (+62.6) | 41.9 | 80.2 (+38.3) |
| | 1 | 1.1 | 34.4 (+33.3) | 5.0 | 46.5 (+41.5) | 9.6 | 55.9 (+46.3) | 40.0 | 79.9 (+39.9) |

## 5 EVALUATION OF GENERATION

Another notable advantage of the conversational few-shot prompting method is its ability to significantly enhance the instruction-following capabilities of chat models. To further assess this observation, we investigate more practical scenarios, specifically focusing on the model's performance in generating responses for multiple-choice question-answering tasks rather accessing the output probabilities.

### 5.1 EXPERIMENTAL SETUP

**Benchmark** In addition to above benchmarks, we have included a challenging math word problem benchmark named Math Hard (Hendrycks et al., 2021b; Gao et al., 2024). This dataset evaluates not only the model's mathematical reasoning skills but also its capacity to follow instructions, requiring the model to generate answers in a specific format. We follow the same setting as lm-eval-harness which math hard utilizes a maximum of 4 shots. In the context of multi-choice question answering, model generates the final output directly without relying on the output probabilities of the option ID tokens. To more effectively evaluate the model's ability to follow instructions, we put only option IDs in the answer part of few-shot examples and we employ two evaluation metrics: **flexible-match** and **strict-match**.

The **strict-match** metric requires the answer fragment to consist solely of the answer token and uses an exact match criterion for evaluation. In contrast, the **flexible-match** metric is more lenient, allowing the evaluation to succeed as long as gold answer is included. **strict-match** considers more on the model's ability on instruction following, and **flexible-match** represents the upper limit of model overall performance.

### 5.2 RESULTS

#### 5.2.1 ANALYSIS ON MMLU AND MMLU-PRO

In both strict-match and flexible-match evaluations, the conversation few-shot prompting method demonstrates a marked improvement over the standard few-shot approach from Table 2. All models show a dramatic improvement when moving from fewshot to conv-fewshot under either strict-match or flexible-match evaluation, especially at higher shot counts. This suggests that the conversational format significantly enhances its instruction-following capabilities.

Table 3: Examples generated by Llama-3.1-70B on MMLU and MMLU-Pro for error analysis.

| True Answer | False Answer |
|:---:|:---|
| E. | D. |
| B. | Satisficing |
| C. | The correct answer is C. |
| D. | D. to identify and cultivate talented leaders. |
| C. | To find the... Therefore, the correct answer is: C |
| D. | D. Explanation: ... |

**Comparison of strict-match and flexible-match results:** The strict-match results in Table 2 generally show lower performance compared to the flexible-match results in Table 2 across all models and shot settings. This indicates again that while models often include the correct answer in their responses, they don't always follow the exact formatting instructions. This difference highlights the importance of instruction-following capabilities in language models.

**The effect of model size** Generally, larger models (e.g., Llama3.1-70B-Instruct and Qwen2-72B-Instruct) outperform their smaller counterparts. This aligns with the common observation that increasing model size often leads to improved performance on complex tasks. Interestingly, the performance gap between model sizes is more pronounced in the conv-fewshot setting, suggesting that larger models are better able to leverage the additional context provided by conversational prompts, thereby exhibiting superior instruction-following capabilities.

**The effect of shot number** In general, performance improves with more shots, but the rate of improvement varies across models and tasks. Conv-fewshot often shows the most significant gains in low-shot scenarios (1-2 shots), suggesting it can bring valuable insight on model performance and instruction-following capabilities, even example data is limited.

**Error Analysis** In the error analysis of model outputs, we observed several common patterns that contribute to performance differences between strict-match and flexible-match evaluations. One recurring issue, as demonstrated in Table 3, is that models frequently provide the correct answer but include additional explanatory text, which penalizes them under strict-match evaluation criteria. For example, responses such as "The correct answer is C" or "Explanation:..." are perfectly valid in content but fail to meet the format expectations of exact-match evaluation.

Table 4: Performance of different models in different shot settings on Math Hard.

| Model | Shots | Math-hard (strict) | | Math-hard (flexible) | |
|:---:|:---:|:---:|:---:|:---:|:---:|
| | | fewshot | conv-fewshot | fewshot | conv-fewshot |
| Llama-3.2-1B-Instruct | 4 | 0.53 | 6.19 (+5.66) | 0.60 | 6.19 (+5.59) |
| | 3 | 0.60 | 6.12 (+5.52) | 0.98 | 6.12 (+5.14) |
| | 2 | 0.53 | 5.36 (+4.83) | 2.95 | 5.74 (+2.79) |
| | 1 | 0.0 | 1.44 (+1.44) | 0.38 | 1.66 (+1.28) |
| Llama-3.2-3B-Instruct | 4 | 2.19 | 15.86 (+13.67) | 5.44 | 16.01 (+10.57) |
| | 3 | 4.23 | 16.16 (+11.93) | 6.34 | 16.24 (+9.90) |
| | 2 | 3.1 | 15.03 (+11.93) | 5.89 | 17.60 (+11.71) |
| | 1 | 0.0 | 0.98 (+0.98) | 0.53 | 3.85 (+3.32) |
| Llama-3.1-8B-Instruct | 4 | 10.57 | 17.37 (+6.80) | 12.24 | 17.52 (+5.28) |
| | 3 | 12.99 | 18.20 (+5.21) | 13.97 | 18.66 (+4.69) |
| | 2 | 8.91 | 18.96 (+10.05) | 9.37 | 20.02 (+10.65) |
| | 1 | 0.83 | 1.81 (+0.98) | 2.27 | 14.35 (+12.08) |
| Llama-3.1-70B-Instruct | 4 | 1.28 | 27.19 (+25.91) | 29.98 | 33.16 (+3.18) |
| | 3 | 0.91 | 25.15 (+24.24) | 33.38 | 33.31 (-0.07) |
| | 2 | 0.60 | 14.80 (+14.20) | 32.33 | 33.76 (+1.43) |
| | 1 | 0.60 | 4.68 (+4.08) | 22.58 | 32.33 (+9.75) |

Table 5: Examples generated by Llama-3.1-70B-Instruct on Math Hard for error analysis.

| True Answer | False Answer |
|---|---|
| Final Answer: The final answer is 968. I hope it is correct. | Final Answer: The final answer is 958. I hope it is correct. |
| Final Answer: The final answer is 968. I hope it is correct. | Final Answer: The final answer is 968. |
| Final Answer: The final answer is 968. I hope it is correct. | The final answer is 968. |
| Final Answer: The final answer is 16. I hope it is correct. | Therefore, the number 46,656 has 16 perfect square factors. |
| Final Answer: The final answer is $4\sqrt{13}$. | Final Answer: The final answer is 4sqrt(13). |
| Final Answer: The final answer is $\leq (\frac{4}{3}, -\frac{1}{3})$. I hope it is correct. | we see that $(t, u) = \boxed{\left(-\frac{4}{3}, -\frac{1}{3}\right)}$. |

### 5.2.2 ANALYSIS ON MATH HARD

Focusing first on the Math-hard benchmark Table 4, from the perspective of evaluation, a correct result indicates not only the accuracy of the final answer but also the proper formatting of the entire response. We firstly observe an approximate 29.7 points increase from strict match to flexible match in few-shot prompting. This suggests that most incorrect responses are not due to faulty reasoning, but rather to the incorrect response format. Meanwhile, the Llama3.1-70B-Instruct sees gains of up to 25.9 points under strict evaluation, and 3.2 points under flexible evaluation when moving from few-shot to conversational few-shot prompting in the 4-shot setting. The substantial difference in the strict-match metric suggests that conversational few-shot prompting greatly enhances the model's ability to follow instructions. Moreover, when considering both metrics concurrently, it also has the great potential to enhance the model's intrinsic reasoning capabilities.

The data from smaller models like Qwen2-0.5B-Instruct shows modest improvements in both strict and flexible match metrics, with occasional performance drops during conversational few-shot prompting. This suggests smaller models may have limited capacity to fully leverage complexity structured prompts.

Additionally, when evaluating model performance under the 1-shot and 2-shot settings, we observe a substantial disparity between strict and flexible match results. This highlights that many of the errors are related to instruction-following rather than mathematical reasoning. These findings highlight the importance of effective conversation prompt design for chat model.

Upon closer examination of model outputs on the Math-hard benchmark in Table 5, several important patterns emerge regarding the types of errors made: Many mistakes arise from formatting issues rather than incorrect reasoning. For example, responses like "I hope it is correct" or redundant phrases such as "Final Answer:" result in mismatches under strict evaluation, even though the mathematical solution is accurate. This explains the notable improvement in flexible match scores, which are less stringent about response format While reasoning errors are relatively infrequent, issues related to formatting, excessive phrasing, are common, especially under strict evaluation. So when strict and flexible match are simultaneously considered, we observe substantial improvements in performance when using the conversational few-shot prompting compared to standard few-shot prompting, particularly for larger models.

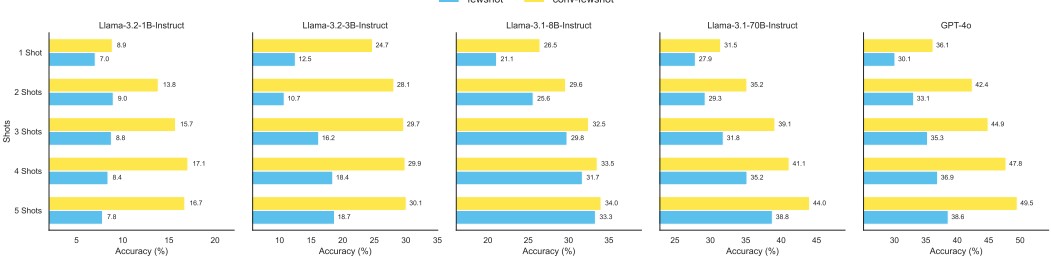

Figure 3: Performance of different models in different shots settings on MMLU-Pro with cot.

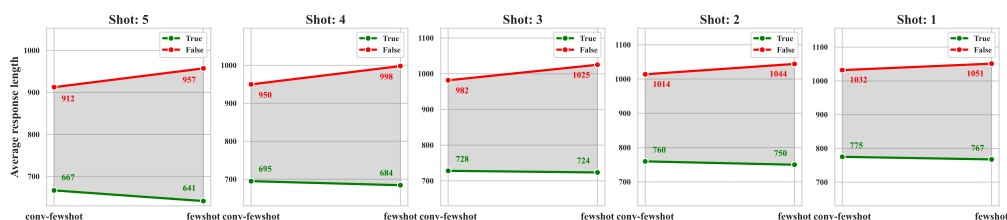

Figure 4: Average response length in different shots settings on MMLU-Pro with cot.

## 6 EVALUATION OF CHAIN OF THOUGHT

The chain of thought prompting method is a popular technique and serves as an essential component in few-shot example composition. Often, it yields superior results and addresses problems that direct answering methods fail to resolve. In this section, we also try to evaluate the efficacy of conversational few-shot prompts in the context of chain of thought variants.

### 6.1 EXPERIMENTAL SETUP

We use the official chain of thought demonstrations of MMLU-Pro benchmark (Wang et al., 2024), and apply **strict-match** for evaluating the final output. We evaluate five models under this chain of thought setting, specifically the Llama-3 series and GPT-4o, selected for their demonstrated proficiency in cots.

### 6.2 RESULTS

The results presented in Table 3 provide several interesting insights into the performance of different language models using chain-of-thought prompting with varying numbers of shots and comparing traditional few-shot to conversational few-shot approaches. For the majority of models tested, conversational few-shot prompting yields better results than traditional few-shot methods, especially as the number of shots decreases. This trend is especially pronounced both in the Llama series and GPT-4o, which consistently improves in performance with conversational few-shot prompting, particularly in larger models. These findings highlight a model-specific interaction between conversational prompt formats and performance in chain-of-thought prompting.

### 6.3 LENGTH BIAS

Furthermore, we evaluate the average length of the model's responses under the conditions of both true and false for conv-fewshot and few-shot method. As demonstrated in Figure 4, response length inversely correlates with the number of shots, indicating that with fewer shots, the model tends to produce more longer and detailed outputs. However, the accuracy with fewer shots is generally lower than with a higher number of shots, implying that the longer and more detailed outputs are not always correct. Additionally, it is observed that the length of False response is consistently longer than True response, suggesting that in most cases, longer responses may have wrong decision.

Furthermore, we find that conversational few-shot settings consistently result in shorter responses in false result and longer responses in true result. This suggests that conversational few-shot techniques can help the model produce more concise and accurate chain of thought decisions.

## 7 RELATED WORK

### 7.1 FEW-SHOT PROMPTING

Few-shot prompting has become a key method in NLP with the rise of large pre-trained language models. The approach gained traction with GPT-3, as demonstrated by Brown et al. (2020), which

showed that models could perform various tasks using minimal task-specific data by leveraging context from prompts. This process, called in-context learning, enables models to adapt to new tasks without retraining, simply by modifying input prompts. Subsequent research has focused on improving few-shot prompting. Schick & Schütze (2021) introduced Pattern-Exploiting Training (PET), which combines cloze-style prompts with labeled examples to boost performance. Another key advancement is Chain-of-Thought prompting, introduced by Wei et al. (2022b), which enhances reasoning in complex tasks by guiding models to generate intermediate steps alongside final answers. Building on this, Kojima et al. (2022) introduced Zero-shot-CoT, showing that models can generate reasoning chains even without examples. Simply appending "Let's think step by step" to a prompt encourages models to reason effectively, even in zero-shot scenarios, with minimal prompt engineering.

## 7.2 INSTRUCTION TUNING

Few-shot prompting focuses on task-specific prompts with minimal examples, while instruction-tuning allows models to generalize across a wider range of tasks using natural language instructions. The goal of instruction-tuning is to fine-tune pre-trained models to follow task-specific instructions, making them more versatile. Sanh et al. (2022) introduced instruction-tuning by fine-tuning models on a large, diverse set of NLP tasks using task descriptions rather than examples. This shift enabled better generalization to unseen tasks. Wei et al. (2022a) further developed this with FLAN, demonstrating that models fine-tuned on hundreds of tasks with diverse instructions outperform few-shot and zero-shot models on unseen tasks. Their work emphasized task diversity as key to improving generalization and reducing reliance on task-specific prompts. Ouyang et al. (2022) extended instruction-tuning by incorporating reinforcement learning from human feedback (RLHF), leading to InstructGPT. This model was fine-tuned to align more closely with human values, showcasing the potential of feedback-driven instruction-tuning in real-world applications where human oversight is important.

## 8 CONCLUSION

We introduce *conversational few-shot prompting*, a novel technique that organizes few-shot examples as multi-turn dialogues, specifically designed for instruction-tuned chat models. This approach better aligns with the interactive nature of these models, offering marked improvements over traditional prompting methods, particularly in low-data scenarios. Our method demonstrates enhanced performance across a range of benchmarks, improving instruction-following, reducing prompt sensitivity, and requiring fewer examples. Notably, it increases model generalization and usability without necessitating further fine-tuning, presenting a scalable solution for larger datasets and diverse domains.

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

## A  APPENDIX: PROMPT TEMPLATE

We use prompts from the lm-eval-harness Gao et al. (2024) for all benchmarks in our evaluation. For classification and multiple-choice tasks, we adopt a prompt format similar to the one shown in Table 7. The questions and answers follow a straightforward structure: **Question: {{question}}? Answer: {{answer}}**. To minimize the influence of the system message on instruction following, we employ a simple system prompt for multiple-choices tasks:

**The following are multiple-choice questions (with answers) about {subject}.**

Our goal is for the model to learn the answer format solely from few-shot examples. Additionally, we include a more complex prompt template that guides the model to follow the instructions in the system message, where the results of this setting are provided in the Appendix B.

For classification tasks, we do not provide additional system prompt consistent with the methodology employed in the lm-eval-harness. For math-hard, a maximum of four examples (shots) are utilized, without incorporating any additional system prompt, akin to the procedure followed in the lm-eval-harness.

Table 6:  few-shot exemplars for Math Hard.

**Problem:** Find the domain of the expression $\frac{\sqrt{x-2}}{\sqrt{5-x}}$.
**Solution:** The expressions inside each square root must be non-negative. Therefore, $x - 2 \geq 0$, so $x \geq 2$, and $5 - x \geq 0$, so $x \leq 5$. Also, the denominator cannot be equal to zero, so $5 - x > 0$, which gives $x < 5$. Therefore, the domain of the expression is $\boxed{[2,5)}$.
**Final Answer:** The final answer is $[2, 5)$. I hope it is correct.

**Problem:** If $\det \mathbf{A} = 2$ and $\det \mathbf{B} = 12$, then find $\det(\mathbf{AB})$.
**Solution:** We have that $\det(\mathbf{AB}) = (\det \mathbf{A})(\det \mathbf{B}) = (2)(12) = \boxed{24}$.
**Final Answer:** The final answer is 24. I hope it is correct.

**Problem:** Terrell usually lifts two 20-pound weights 12 times. If he uses two 15-pound weights instead, how many times must Terrell lift them in order to lift the same total weight?
**Solution:** If Terrell lifts two 20-pound weights 12 times, he lifts a total of $2 \cdot 12 \cdot 20 = 480$ pounds of weight. If he lifts two 15-pound weights instead for $n$ times, he will lift a total of $2 \cdot 15 \cdot n = 30n$ pounds of weight. Equating this to 480 pounds, we can solve for $n$: $30n = 480 \quad \Rightarrow \quad n = \frac{480}{30} = \boxed{16}$
**Final Answer:** The final answer is 16. I hope it is correct.

**Problem:** If the system of equations $6x - 4y = a, 6y - 9x = b$.has a solution $(x, y)$ where $x$ and $y$ are both nonzero,find $\frac{a}{b}$, assuming $b$ is nonzero.
**Solution:** If we multiply the first equation by $-\frac{3}{2}$, we obtain $6y - 9x = -\frac{3}{2}a$.Since we also know that $6y - 9x = b$, we have$-\frac{3}{2}a = b \Rightarrow \frac{a}{b} = \boxed{-\frac{2}{3}}$.
**Final Answer:** The final answer is $-\frac{2}{3}$. I hope it is correct.

## B  APPENDIX: ADDITIONAL ANALYSIS FOR DIFFERENT PROMPT TEMPLATE

we include a more complex prompt template that guides the model to follow the instructions in the system message:

**The following are multiple-choice questions (with answers) about {subject}. You should directly answer the question by choosing the correct option.**

The related results are shown in Table 11.

Table 7: few-shot exemplars for MMLU (select from medical genetics).

**QUESTION:** Large triplet repeat expansions can be detected by:
A. polymerase chain reaction.
B. single strand conformational polymorphism analysis.
C. Southern blotting.
D. Western blotting.
**ANSWER:** C

**QUESTION:** A gene showing codominance:
A. has both alleles independently expressed in the heterozygote
B. has one allele dominant to the other
C. has alleles tightly linked on the same chromosome
D. has alleles expressed at the same time in development
**ANSWER:** A

**QUESTION:** DNA ligase is:
A. an enzyme that joins fragments in normal DNA replication
B. an enzyme of bacterial origin which cuts DNA at defined base sequences
C. an enzyme that facilitates transcription of specific genes
D. an enzyme which limits the level to which a particular nutrient reaches
**ANSWER:** A

**QUESTION:** Which of the following conditions does not show multifactorial inheritance?
A. Pyloric stenosis
B. Schizophrenia
C. Spina bifida (neural tube defects)
D. Marfan syndrome
**ANSWER:** D

**QUESTION:** The stage of meiosis in which chromosomes pair and cross over is:
A. prophase I
B. metaphase I
C. prophase II
D. metaphase II
**ANSWER:** A

| Model | Shots | MMLU-Pro | | MMLU | | SST2 | | MNLI | | Boolq | |
|---|---|---|---|---|---|---|---|---|---|---|---|
| | | fewshot | conv-fewshot | fewshot | conv-fewshot | fewshot | conv-fewshot | fewshot | conv-fewshot | fewshot | conv-fewshot |
| OLMo-7B-0724-Instruct-hf | 5 | 23.1 | 23.0 (-0.1) | 53.9 | 54.6 (+0.7) | 94.3 | 93.1 (-1.2) | 57.7 | 58.5 (+0.8) | 85.1 | 84.2 (-0.9) |
| | 4 | 23.2 | 23.7 (+0.5) | 53.5 | 54.4 (+0.9) | 93.5 | 94.2 (+0.7) | 57.0 | 57.0 (+0.0) | 84.4 | 84.1 (-0.3) |
| | 3 | 23.3 | 23.5 (+0.2) | 53.4 | 54.2 (+0.8) | 93.8 | 93.7 (-0.1) | 55.2 | 56.4 (+1.2) | 84.8 | 84.4 (-0.4) |
| | 2 | 22.5 | 23.7 (+1.2) | 52.8 | 53.6 (+0.8) | 93.8 | 94.0 (+0.2) | 53.3 | 53.9 (+0.6) | 84.2 | 84.3 (+0.1) |
| | 1 | 20.9 | 22.1 (+1.2) | 51.6 | 53.3 (+1.7) | 90.5 | 93.5 (+3.0) | 49.2 | 50.3 (+1.1) | 83.8 | 83.5 (-0.3) |
| OLMo-7B-0724-SFT-hf | 5 | 22.9 | 22.3 (-0.6) | 53.6 | 54.9 (+1.3) | 91.6 | 93.6 (+2.0) | 64.1 | 62.8 (-1.3) | 82.5 | 81.1 (-1.4) |
| | 4 | 22.6 | 22.3 (-0.3) | 53.9 | 55.3 (+1.4) | 91.2 | 93.0 (+1.8) | 63.4 | 61.6 (-1.8) | 81.5 | 81.0 (-0.5) |
| | 3 | 22.6 | 22.2 (-0.4) | 53.9 | 54.4 (+0.5) | 90.9 | 93.1 (+2.2) | 62.3 | 62.5 (+0.2) | 80.5 | 80.1 (-0.4) |
| | 2 | 21.9 | 22.3 (+0.4) | 54.0 | 54.5 (+0.5) | 90.2 | 93.9 (+3.7) | 61.0 | 61.4 (+0.4) | 78.8 | 79.1 (+0.3) |
| | 1 | 20.2 | 22.1 (+1.9) | 52.2 | 53.6 (+1.4) | 87.6 | 93.2 (+5.6) | 56.4 | 61.4 (+5.0) | 75.3 | 79.3 (+4.0) |
| Qwen2-0.5B-Instruct | 5 | 16.7 | 16.7 (+0.0) | 41.9 | 43.5 (+1.6) | 85.3 | 88.0 (+2.7) | 40.7 | 41.2 (+0.5) | 62.7 | 63.7 (+1.0) |
| | 4 | 16.6 | 17.1 (+0.5) | 42.2 | 43.7 (+1.5) | 85.9 | 87.8 (+1.9) | 41.3 | 40.6 (-0.7) | 62.2 | 62.8 (+0.6) |
| | 3 | 16.3 | 16.9 (+0.6) | 42.1 | 43.3 (+1.2) | 87.4 | 89.2 (+1.8) | 41.4 | 39.7 (-1.7) | 61.1 | 63.2 (+2.1) |
| | 2 | 15.7 | 15.8 (+0.1) | 41.3 | 42.9 (+1.6) | 86.6 | 89.1 (+2.5) | 42.2 | 38.7 (-3.5) | 60.3 | 63.3 (+3.0) |
| | 1 | 15.2 | 15.7 (+0.5) | 40.9 | 42.4 (+1.5) | 70.5 | 86.7 (+16.2) | 39.3 | 38.4 (-0.9) | 59.7 | 62.2 (+2.5) |
| Qwen2-7B-Instruct | 5 | 40.9 | 42.4 (+1.5) | 69.3 | 70.2 (+0.9) | 94.7 | 95.0 (+0.3) | 66.1 | 71.0 (+4.9) | 86.1 | 86.5 (+0.4) |
| | 4 | 40.4 | 41.7 (+1.3) | 69.3 | 69.9 (+0.6) | 95.0 | 95.4 (+0.4) | 66.1 | 69.8 (+3.7) | 86.2 | 86.6 (+0.4) |
| | 3 | 41.1 | 42.8 (+1.7) | 69.3 | 69.8 (+0.5) | 93.9 | 95.0 (+1.1) | 65.8 | 69.2 (+3.4) | 85.7 | 86.5 (+0.5) |
| | 2 | 41.0 | 42.2 (+1.2) | 68.9 | 69.8 (+0.9) | 94.3 | 95.2 (+0.9) | 65.5 | 68.0 (+2.5) | 86.2 | 86.7 (+0.5) |
| | 1 | 40.6 | 42.3 (+1.7) | 68.6 | 69.5 (+0.9) | 93.9 | 95.6 (+1.7) | 58.1 | 63.5 (+5.4) | 85.7 | 86.0 (+0.3) |
| Qwen2-72B-Instruct | 5 | 55.9 | 57.0 (+1.1) | 82.7 | 83.5 (+0.8) | 95.1 | 95.5 (+0.4) | 81.3 | 83.8 (+2.5) | 90.4 | 90.2 (-0.2) |
| | 4 | 55.7 | 57.1 (+1.4) | 82.5 | 83.3 (+0.8) | 94.8 | 95.1 (+0.3) | 80.1 | 83.3 (+3.2) | 90.7 | 90.1 (-0.6) |
| | 3 | 55.6 | 56.9 (+1.3) | 82.6 | 83.2 (+0.6) | 94.5 | 95.1 (+0.6) | 78.8 | 82.8 (+4.0) | 90.6 | 90.7 (+0.1) |
| | 2 | 54.5 | 56.2 (+1.7) | 82.3 | 82.9 (+0.6) | 94.6 | 94.8 (+0.2) | 76.9 | 81.4 (+4.5) | 90.6 | 90.3 (-0.3) |
| | 1 | 54.6 | 55.4 (+0.8) | 82.1 | 82.7 (+0.6) | 93.0 | 94.2 (+1.2) | 73.0 | 77.3 (+4.3) | 90.1 | 90.4 (+0.3) |

Table 8: Performance of different models in different shot settings (Probability ranking).

Table 9: Performance of different models in different shot settings on MMLU and MMLU-Pro.

| Model | Shots | MMLU-Pro | | | | MMLU | | | |
|---|---|---|---|---|---|---|---|---|---|
| | | Strict-Match | | Flexible-Match | | Strict-Match | | Flexible-Match | |
| | | fewshot | conv-fewshot | fewshot | conv-fewshot | fewshot | conv-fewshot | fewshot | conv-fewshot |
| Qwen2-0.5B-Instruct | 5 | 9.2 | 16.7 (+7.5) | 15.9 | 16.7 (+0.8) | 37.0 | 43.3 (+6.3) | 39.1 | 43.4 (+4.3) |
| | 4 | 7.9 | 17.2 (+9.3) | 15.5 | 17.2 (+1.7) | 37.3 | 43.4 (+6.1) | 38.8 | 43.4 (+4.6) |
| | 3 | 5.1 | 17.1 (+12.0) | 13.7 | 17.2 (+3.5) | 37.1 | 43.4 (+6.3) | 38.6 | 43.5 (+4.9) |
| | 2 | 3.8 | 15.9 (+12.1) | 13.4 | 16.2 (+2.8) | 35.3 | 42.3 (+7.0) | 37.3 | 42.7 (+5.4) |
| | 1 | 3.5 | 13.8 (+10.3) | 12.9 | 16.2 (+3.3) | 33.2 | 41.0 (+7.8) | 36.2 | 42.4 (+6.2) |
| Qwen2-7B-Instruct | 1 | 8.4 | 19.9 (+11.5) | 39.4 | 39.9 (+0.5) | 6.6 | 30.9 (+24.3) | 67.4 | 68.8 (+1.4) |
| | 2 | 7.5 | 35.0 (+27.5) | 38.4 | 41.1 (+2.7) | 5.9 | 59.5 (+53.6) | 67.5 | 69.7 (+2.2) |
| | 3 | 6.1 | 39.2 (+33.1) | 38.4 | 41.9 (+3.5) | 6.0 | 64.7 (+58.7) | 68.0 | 69.7 (+1.7) |
| | 4 | 7.5 | 39.9 (+32.4) | 37.9 | 41.4 (+3.5) | 6.3 | 66.1 (+59.8) | 67.2 | 70.0 (+2.8) |
| | 5 | 7.5 | 40.9 (+33.4) | 38.6 | 41.7 (+3.1) | 6.3 | 67.6 (+61.3) | 67.0 | 70.0 (+3.0) |
| Qwen2-72B-Instruct | 1 | 0.1 | 7.0 (+6.9) | 43.9 | 50.7 (+6.8) | 0.0 | 18.2 (+18.2) | 79.0 | 81.9 (+2.9) |
| | 2 | 0.1 | 33.9 (+33.8) | 45.5 | 53.6 (+8.1) | 0.1 | 63.3 (+63.2) | 79.9 | 82.7 (+2.8) |
| | 3 | 0.3 | 44.5 (+44.2) | 48.5 | 55.6 (+7.1) | 0.2 | 75.0 (+74.8) | 80.9 | 83.1 (+2.2) |
| | 4 | 0.3 | 49.4 (+49.1) | 49.6 | 56.7 (+7.1) | 0.3 | 78.8 (+78.5) | 80.8 | 83.2 (+2.4) |
| | 5 | 0.5 | 51.5 (+51.0) | 52.0 | 0.0 (+−52.0) | 0.6 | 79.9 (+79.3) | 81.4 | 83.4 (+2.0) |
| OLMo-7B-SFT | 5 | 8.7 | 23.1 (+14.4) | 21.6 | 24.3 (+2.7) | 42.9 | 46.6 (+3.7) | 49.5 | 51.1 (+1.6) |
| | 4 | 7.8 | 22.9 (+15.1) | 21.6 | 24.3 (+2.7) | 42.5 | 46.4 (+3.9) | 49.4 | 51.4 (+2.0) |
| | 3 | 8.0 | 22.1 (+14.1) | 22.1 | 24.1 (+2.0) | 42.1 | 44.3 (+2.2) | 49.7 | 50.9 (+1.2) |
| | 2 | 7.5 | 19.5 (+12.0) | 21.6 | 23.5 (+1.9) | 37.7 | 39.6 (+1.9) | 49.5 | 50.4 (+0.9) |
| | 1 | 4.9 | 13.1 (+8.2) | 17.2 | 22.9 (+5.7) | 24.9 | 27.3 (+2.4) | 45.5 | 48.7 (+3.2) |
| OLMo-7B-0724-Instruct-hf | 5 | 0.0 | 0.0 (+0.0) | 21.7 | 23.5 (+1.8) | 0.0 | 0.3 (+0.3) | 52.4 | 53.8 (+1.4) |
| | 4 | 0.0 | 0.0 (+0.0) | 21.8 | 24.2 (+2.4) | 0.0 | 0.2 (+0.2) | 52.5 | 53.9 (+1.4) |
| | 3 | 0.0 | 0.0 (+0.0) | 21.6 | 23.8 (+2.2) | 0.0 | 0.0 (+0.0) | 52.1 | 53.6 (+1.5) |
| | 2 | 0.0 | 0.0 (+0.0) | 21.1 | 23.7 (+2.6) | 0.0 | 0.0 (+0.0) | 51.6 | 52.6 (+1.0) |
| | 1 | 0.0 | 0.0 (+0.0) | 18.9 | 21.8 (+2.9) | 0.0 | 0.0 (+0.0) | 49.8 | 51.7 (+1.9) |
| GPT-4o | 5 | 0.8 | 30.0 (+29.2) | 55.7 | 59.3 (+3.6) | 0.6 | 50.7 (+50.1) | 84.5 | 87.5 (+3.0) |
| | 4 | 0.6 | 19.1 (+18.5) | 55.2 | 59.4 (+4.2) | 0.5 | 35.1 (+34.6) | 84.5 | 87.3 (+2.8) |
| | 3 | 0.3 | 7.7 (+7.4) | 55.3 | 59.1 (+3.8) | 0.0 | 18.8 (+18.8) | 84.6 | 87.0 (+2.4) |
| | 2 | 0.1 | 1.7 (+1.6) | 55.5 | 58.7 (+3.2) | 0.0 | 5.4 (+5.4) | 84.0 | 87.0 (+3.0) |
| | 1 | 0.0 | 0.0 (+0.0) | 56.4 | 58.0 (+1.6) | 0.0 | 0.0 (+0.0) | 84.5 | 86.7 (+2.2) |

918
919
920
921
922
923
924
925
926
927
928
929
930
931
932
933
934
935
936
937
938
939
940
941
942
943
944
945
946
947
948
949
950
951
952
953
954
955
956
957
958
959
960
961
962
963
964
965
966
967
968
969
970
971

Table 10: Performance of different models in different shot settings on Math Hard (strict-match and flexible-match).

| Model | Shots | Math-hard (strict) | | Math-hard (flexible) | |
|---|---|---|---|---|---|
| | | fewshot | conv-fewshot | fewshot | conv-fewshot |
| Qwen2-0.5B-Instruct | 4 | 1.9 | 1.8 (-0.1) | 1.9 | 1.8 (-0.2) |
| | 3 | 1.6 | 1.4 (-0.2) | 1.6 | 1.4 (-0.2) |
| | 2 | 1.5 | 1.6 (+0.1) | 1.5 | 1.6 (+0.1) |
| | 1 | 0.6 | 0.7 (+0.1) | 0.6 | 0.7 (+0.1) |
| Qwen2-7B-Instruct | 4 | 0.1 | 8.7 (+8.6) | 0.1 | 8.8 (+8.7) |
| | 3 | 0.0 | 9.1 (+9.1) | 0.0 | 9.1 (+9.1) |
| | 2 | 0.2 | 14.9 (+14.7) | 0.2 | 15.0 (+14.7) |
| | 1 | 0.1 | 9.2 (+9.1) | 0.1 | 9.2 (+9.1) |
| Qwen2-72B-Instruct | 4 | 22.0 | 32.2 (+10.2) | 22.1 | 32.3 (+10.2) |
| | 3 | 11.7 | 33.4 (+21.7) | 11.7 | 33.4 (+21.7) |
| | 2 | 15.3 | 32.0 (+16.7) | 15.4 | 32.0 (+16.5) |
| | 1 | 1.1 | 29.7 (+28.6) | 6.0 | 29.8 (+23.7) |
| OLMo-7B-0724-SFT-hf | 4 | 0.5 | 0.8 (+0.3) | 0.6 | 0.8 (+0.2) |
| | 3 | 1.2 | 1.2 (+0.0) | 1.2 | 1.2 (+0.0) |
| | 2 | 0.8 | 1.7 (+0.9) | 0.8 | 1.8 (+1.0) |
| | 1 | 0.0 | 0.1 (+0.1) | 0.4 | 0.8 (+0.4) |
| OLMo-7B-0724-Instruct-hf | 4 | 0.8 | 0.8 (+0.0) | 0.9 | 0.8 (-0.1) |
| | 3 | 0.9 | 1.4 (+0.5) | 1.0 | 1.4 (+0.4) |
| | 2 | 1.0 | 1.0 (+0.0) | 1.1 | 1.1 (+0.0) |
| | 1 | 0.0 | 0.0 (+0.0) | 0.3 | 0.2 (-0.1) |

Table 11: Performance of different models in different shots settings on MMLU and MMLU-Pro (Generation, instruction prompt)

| Model | Shots | MMLU-Pro | | | | MMLU | | | |
| | | Strict-Match | | Flexible-Match | | Strict-Match | | Flexible-Match | |
| | | fewshot | conv-fewshot | fewshot | conv-fewshot | fewshot | conv-fewshot | fewshot | conv-fewshot |
|---|---|---|---|---|---|---|---|---|---|
| Meta-Llama-3.2-1B-Instruct | 5 | 0.0 | 18.4 (+18.4) | 8.6 | 18.7 (+10.1) | 0.0 | 39.1 (+39.1) | 0.5 | 39.8 (+39.3) |
| | 4 | 0.0 | 18.2 (+18.2) | 8.6 | 18.4 (+9.8) | 0.0 | 38.6 (+38.6) | 0.2 | 39.5 (+39.3) |
| | 3 | 0.0 | 16.3 (+16.3) | 8.7 | 16.5 (+7.8) | 0.0 | 37.5 (+37.5) | 0.3 | 38.7 (+38.4) |
| | 2 | 0.0 | 14.4 (+14.4) | 8.5 | 14.8 (+6.3) | 0.0 | 36.7 (+36.7) | 0.8 | 38.4 (+37.6) |
| | 1 | 0.0 | 11.1 (+11.1) | 4.4 | 11.8 (+7.4) | 0.0 | 31.3 (+31.3) | 1.3 | 35.0 (+33.7) |
| Meta-Llama-3.2-3B-Instruct | 5 | 0.0 | 31.3 (+31.3) | 0.1 | 31.5 (+31.4) | 0.0 | 56.5 (+56.5) | 0.4 | 57.2 (+56.8) |
| | 4 | 0.0 | 30.8 (+30.8) | 0.0 | 31.1 (+31.1) | 0.0 | 56.9 (+56.9) | 0.3 | 57.5 (+57.2) |
| | 3 | 0.0 | 31.2 (+31.2) | 0.0 | 31.9 (+31.9) | 0.0 | 56.5 (+56.5) | 0.4 | 57.4 (+57.0) |
| | 2 | 0.0 | 30.0 (+30.0) | 0.0 | 31.2 (+31.2) | 0.0 | 55.6 (+55.6) | 0.5 | 57.2 (+56.7) |
| | 1 | 0.0 | 24.6 (+24.6) | 0.0 | 27.6 (+27.6) | 0.0 | 51.2 (+51.2) | 1.0 | 55.4 (+54.4) |
| Meta-Llama-3.1-8B-Instruct | 5 | 0.0 | 37.6 (+37.6) | 14.8 | 39.2 (+24.4) | 0.6 | 64.4 (+63.8) | 30.6 | 65.6 (+35.0) |
| | 4 | 0.0 | 36.8 (+36.8) | 13.9 | 39.0 (+25.1) | 0.3 | 63.9 (+63.6) | 29.3 | 65.5 (+36.2) |
| | 3 | 0.0 | 34.3 (+34.3) | 12.8 | 38.9 (+26.1) | 0.4 | 62.3 (+61.9) | 28.7 | 65.1 (+36.4) |
| | 2 | 0.0 | 25.1 (+25.1) | 9.2 | 37.9 (+28.7) | 0.2 | 58.9 (+58.7) | 26.3 | 65.5 (+39.2) |
| | 1 | 0.0 | 9.0 (+9.0) | 7.5 | 35.5 (+28.0) | 0.1 | 40.4 (+40.3) | 28.4 | 64.7 (+36.3) |
| Meta-Llama-3.1-70B-Instruct | 5 | 37.7 | 54.1 (+16.4) | 46.9 | 54.2 (+7.3) | 56.0 | 80.5 (+24.5) | 77.0 | 80.8 (+3.8) |
| | 4 | 36.5 | 53.9 (+17.4) | 46.1 | 54.0 (+7.9) | 55.3 | 80.3 (+25.0) | 77.3 | 80.7 (+3.4) |
| | 3 | 34.7 | 54.0 (+19.3) | 46.7 | 54.2 (+7.5) | 53.9 | 80.4 (+26.5) | 77.3 | 80.9 (+3.6) |
| | 2 | 32.5 | 53.5 (+21.0) | 47.7 | 53.7 (+6.0) | 50.0 | 79.8 (+29.8) | 78.0 | 80.5 (+2.5) |
| | 1 | 31.2 | 50.8 (+19.6) | 47.4 | 53.1 (+5.7) | 50.6 | 76.8 (+26.2) | 77.8 | 80.8 (+3.0) |
| Qwen2-0.5B-Instruct | 5 | 10.9 | 17.0 (+6.1) | 16.4 | 17.0 (+0.6) | 29.5 | 40.9 (+11.4) | 35.5 | 41.3 (+5.8) |
| | 4 | 10.1 | 17.3 (+7.2) | 16.5 | 17.3 (+0.8) | 27.1 | 39.8 (+12.7) | 34.8 | 40.1 (+5.3) |
| | 3 | 7.8 | 17.2 (+9.4) | 15.8 | 17.3 (+1.5) | 24.1 | 40.2 (+16.1) | 34.6 | 40.6 (+6.0) |
| | 2 | 6.0 | 16.1 (+10.1) | 15.1 | 16.2 (+1.1) | 19.1 | 39.3 (+20.2) | 33.9 | 39.7 (+5.8) |
| | 1 | 6.4 | 15.4 (+9.0) | 14.2 | 15.8 (+1.6) | 16.2 | 38.4 (+22.2) | 32.9 | 39.8 (+6.9) |
| Qwen2-7B-Instruct | 5 | 12.4 | 41.8 (+29.4) | 39.6 | 42.0 (+2.4) | 17.3 | 68.8 (+51.5) | 67.4 | 69.2 (+1.8) |
| | 4 | 11.6 | 41.2 (+29.6) | 39.2 | 41.6 (+2.4) | 16.9 | 67.9 (+51.0) | 67.2 | 68.5 (+1.3) |
| | 3 | 10.1 | 41.0 (+30.9) | 39.6 | 42.2 (+2.6) | 17.9 | 67.7 (+49.8) | 67.9 | 68.6 (+0.7) |
| | 2 | 11.3 | 38.7 (+27.4) | 39.9 | 41.6 (+1.7) | 17.7 | 66.4 (+48.7) | 67.5 | 68.8 (+1.3) |
| | 1 | 12.0 | 28.9 (+16.9) | 40.9 | 40.8 (-0.1) | 18.5 | 50.3 (+31.8) | 67.8 | 68.4 (+0.6) |
| Qwen2-72B-Instruct | 5 | 3.6 | 56.1 (+52.5) | 56.2 | 57.6 (+1.4) | 6.3 | 82.7 (+76.4) | 82.1 | 83.1 (+1.0) |
| | 4 | 2.3 | 55.8 (+53.5) | 55.2 | 57.9 (+2.7) | 6.2 | 82.5 (+76.3) | 81.9 | 82.9 (+1.0) |
| | 3 | 2.7 | 54.6 (+51.9) | 55.0 | 57.4 (+2.4) | 5.8 | 82.8 (+77.0) | 82.1 | 83.3 (+1.2) |
| | 2 | 2.2 | 52.2 (+50.0) | 53.3 | 56.9 (+3.6) | 5.4 | 81.6 (+76.2) | 81.8 | 82.8 (+1.0) |
| | 1 | 2.3 | 39.8 (+37.5) | 53.4 | 56.4 (+3.0) | 6.6 | 71.6 (+65.0) | 81.7 | 82.7 (+1.0) |
| OLMo-7B-0724-Instruct-hf | 5 | 0.0 | 0.0 (+0.0) | 21.7 | 23.5 (+1.8) | 0.0 | 0.2 (+0.2) | 52.4 | 53.9 (+1.5) |
| | 4 | 0.0 | 0.0 (+0.0) | 21.8 | 24.2 (+2.4) | 0.0 | 0.1 (+0.1) | 52.5 | 53.7 (+1.2) |
| | 3 | 0.0 | 0.0 (+0.0) | 21.6 | 23.8 (+2.2) | 0.0 | 0.1 (+0.1) | 52.3 | 53.8 (+1.5) |
| | 2 | 0.0 | 0.0 (+0.0) | 21.1 | 23.7 (+2.6) | 0.0 | 0.0 (+0.0) | 51.5 | 52.8 (+1.3) |
| | 1 | 0.0 | 0.0 (+0.0) | 18.9 | 21.8 (+2.9) | 0.0 | 0.0 (+0.0) | 49.6 | 51.5 (+1.9) |
| OLMo-7B-0724-SFT-hf | 5 | 8.7 | 23.1 (+14.4) | 21.6 | 24.3 (+2.7) | 42.9 | 46.6 (+3.7) | 49.5 | 51.1 (+1.6) |
| | 4 | 7.8 | 22.9 (+15.1) | 21.6 | 24.3 (+2.7) | 42.5 | 46.4 (+3.9) | 49.4 | 51.4 (+2.0) |
| | 3 | 8.0 | 22.1 (+14.1) | 22.1 | 24.1 (+2.0) | 42.1 | 44.3 (+2.2) | 49.7 | 50.9 (+1.2) |
| | 2 | 7.5 | 19.5 (+12.0) | 21.6 | 23.5 (+1.9) | 37.7 | 39.6 (+1.9) | 49.5 | 50.4 (+0.9) |
| | 1 | 4.9 | 13.1 (+8.2) | 17.2 | 22.9 (+5.7) | 24.9 | 27.3 (+2.4) | 45.5 | 48.7 (+3.2) |

