# OpenReview forum: "Conversational Few-Shot Prompting: Rethinking Few-Shot Prompting for Chat Language Model"
_ICLR.cc/2025/Conference — Submitted to ICLR 2025_

### Official Review · Reviewer_VjQ9 · 2024-11-04

**Soundness:** 1
**Presentation:** 2
**Contribution:** 2
**Rating:** 3
**Confidence:** 5

**Summary:**

The paper proposes Conversational Few-Shot Prompting for chat models, transforming few-shot examples into multi-turn dialogues to align with conversational model dynamics. Conversational Few-Shot Prompting improves instruction-following and generalization, outperforming traditional few-shot prompts in low-shot scenarios across benchmarks without additional fine-tuning, enhancing task performance and reducing prompt sensitivity.

**Strengths:**

1. Revises few-shot prompting by structuring examples as multi-turn dialogues, aligning better with conversational models.
2. Demonstrates enhanced performance across multiple tasks in various few-shot configurations and ability to follow complex instructions.

**Weaknesses:**

1. The approach relies on prompt engineering without a clear explanation of why the multi-turn structure improves overall performance or enhances instruction-following capabilities. Why would putting few-shot exemplars in the chat be beneficial? Is there any potential mechanism behind the scenes?
2. The paper offers limited insight into error cases or scenarios where the method may perform poorly, which would provide a clearer understanding of its robustness across different applications.

**Questions:**

1. In instruction-following tasks, where are the instructions placed within the prompt in the conversational format? Could the authors provide an example comparing how instructions are incorporated in this approach versus traditional instruction-following formats?
2. Are there specific tasks or model architectures where this approach might be less effective? Have the authors considered edge cases where the conversational context may not provide improvements or could even hinder performance?
3. How does this approach perform relative to other prompting techniques (such as CoT+few shot vs conv-fewshot without CoT), particularly in complex reasoning scenarios?
4. What types of errors are most frequent with this method, particularly across different task types or complexity levels? Are there specific failure cases that occur more often in certain scenarios (e.g., instruction-following vs. complex reasoning tasks)? Could a detailed error analysis, possibly by task type or difficulty level, reveal areas for potential improvement or further refinement?
5. What are the decoding parameters for the baseline prompts and the conversational prompts? Any randomness is included during the decoding steps? How many runs are conducted for evaluating each task?

---

> ### Comment · Reviewer_VjQ9 · 2024-11-26
>
> Thank you for your explanation. Few-shot learning was initially developed before the advent of instruction-tuned models, which explains the absence of role-specific formatting in early approaches. With the emergence of instruction-tuned models, incorporating user and assistant tokens in prompts has become a natural progression, aligning with the supervised fine-tuning (SFT) format. This methodology has been previously discussed in various sources, such as Microsoft's documentation on the following examples:
>
> [Few shot learning with ChatML](https://learn.microsoft.com/en-us/azure/ai-services/openai/how-to/chat-markup-language#few-shot-learning-with-chatml):
> ```
> <|im_start|>system
> Assistant is an intelligent chatbot designed to help users answer their tax related questions.
> <|im_end|>
> <|im_start|>user
> When do I need to file my taxes by?
> <|im_end|>
> <|im_start|>assistant
> In 2023, you will need to file your taxes by April 18th. The date falls after the usual April 15th deadline because April 15th falls on a Saturday in 2023. For more details, see https://www.irs.gov/filing/individuals/when-to-file
> <|im_end|>
> <|im_start|>user
> How can I check the status of my tax refund?
> <|im_end|>
> <|im_start|>assistant
> You can check the status of your tax refund by visiting https://www.irs.gov/refunds
> <|im_end|>
> ```
>
> [Few-shot learning with chat completion](https://learn.microsoft.com/en-us/azure/ai-services/openai/how-to/chatgpt?tabs=python-new#few-shot-learning-with-chat-completion):
> ```
> {"role": "system", "content": "Assistant is an intelligent chatbot designed to help users answer their tax related questions. "},
> {"role": "user", "content": "When do I need to file my taxes by?"},
> {"role": "assistant", "content": "In 2023, you will need to file your taxes by April 18th. The date falls after the usual April 15th deadline because April 15th falls on a Saturday in 2023. For more details, see https://www.irs.gov/filing/individuals/when-to-file."},
> {"role": "user", "content": "How can I check the status of my tax refund?"},
> {"role": "assistant", "content": "You can check the status of your tax refund by visiting https://www.irs.gov/refunds"}
> ```
>
>
> Given this prior discussion and the natural progression to SFT-compatible formatting, I will maintain my current evaluation scores.

---

> ### Author Response · Authors · 2024-11-27
> **Thanks for your responses.**
>
> Thank you for your thorough feedback and for bringing these references to our attention.
>
> We acknowledge the articles you mentioned, published on August 29 and September 5, 2024. These blogs emerged concurrent with our research timeline, and we did not consider them when doing our work (**According to the ICLR reviewer guidelines,** works published within the last four months of DDL are considered contemporaneous. DDL was 1 October, and four months ago was 1 June).
>
> While these blog posts present valuable insights, they are currently in their preliminary form and have not yet been developed into peer-reviewed academic papers or undergone rigorous scholarly discourse.
>
> We will certainly include these references in our "Related Works" section to provide a comprehensive overview of the current landscape.
> We believe our conversational few-shot prompting will become the standard method in few-shot learning.
>
> We appreciate your diligence in ensuring proper attribution and would be happy to address any additional questions or concerns you may have.

---

> > ### Comment · Reviewer_VjQ9 · 2024-12-03
> >
> > The authors are encouraged to conduct a more thorough literature review before publication. The previously mentioned links were identified through a quick search on few-shot prompting. Similarly, another brief search revealed additional relevant references that appear to have been overlooked.
> >
> > For example, the OpenAI API platform documentation explicitly mentions this type of prompting strategy in their guide: [OpenAI Prompt Engineering Guide](https://platform.openai.com/docs/guides/prompt-engineering#tactic-provide-examples).
> >
> > Additionally, Meta Llama’s GitHub repositories include notebooks titled "Prompt Engineering with Llama 3" ([Llama 3 Notebook](https://github.com/meta-llama/llama-recipes/blob/aab327c2feeae906e5dea9d472bb72ccc76d569b/recipes/quickstart/Prompt_Engineering_with_Llama_3.ipynb), published eight months ago) and "Prompt Engineering with Llama 2" ([Llama 2 Notebook](https://github.com/meta-llama/llama-recipes/blob/147aaa29bcf1e21f965e4530e697268dea132607/examples/Prompt_Engineering_with_Llama_2.ipynb), published on January 17, 2024). Both illustrate few-shot prompting formats. For instance, the following code snippet from the Llama notebooks demonstrates how a sentiment classifier could implement few-shot prompting:
> >
> > ```python
> > def sentiment(text):
> >     response = chat_completion(messages=[
> >         user("You are a sentiment classifier. For each message, give the percentage of positive/neutral/negative."),
> >         user("I liked it"),
> >         assistant("70% positive 30% neutral 0% negative"),
> >         user("It could be better"),
> >         assistant("0% positive 50% neutral 50% negative"),
> >         user("It's fine"),
> >         assistant("25% positive 50% neutral 25% negative"),
> >         user(text),
> >     ])
> >     return response
> >
> > def print_sentiment(text):
> >     print(f'INPUT: {text}')
> >     print(sentiment(text))
> >
> > print_sentiment("I thought it was okay")
> > # Likely to return a balanced mix of positive, neutral, and negative
> > print_sentiment("I loved it!")
> > # Likely to return 100% positive
> > print_sentiment("Terrible service 0/10")
> > # Likely to return 100% negative
> > ```
> >
> >
> > This demonstrates that few-shot prompting techniques, similar to those proposed in the paper, have already been explored and implemented. The authors are encouraged to incorporate such references into their discussion to clarify their work's novelty and contributions more effectively.

---

### Official Review · Reviewer_bHx3 · 2024-11-04

**Soundness:** 2
**Presentation:** 2
**Contribution:** 2
**Rating:** 5
**Confidence:** 3

**Summary:**

This paper addresses the gap in applying in-context learning (or few-shot learning) to instruction-tuned chat models, like ChatGPT, which has been underexplored. It introduces a "conversational few-shot prompting" approach that presents few-shot examples as multi-turn interactions between the user and assistant, rather than a single prompt. This conversational format aligns with the interactive nature of chat models, enhancing their ability to follow instructions and generalize across tasks. Experimental results show that this method improves performance, especially in low-shot scenarios, offering a flexible and robust way to use few-shot examples without additional fine-tuning. This approach reduces prompt sensitivity and opens up potential for diverse applications.

**Strengths:**

- This paper presents a new few-shot prompting method optimized for chat-based, instruction-tuned large language models (LLMs).
- The authors conduct extensive experiments with popular open-source LLMs, including the LLaMA series and Qwen series, across various benchmarks, varying the number of few-shot examples.

**Weaknesses:**

- In my opinion, this technique lacks technical novelty, as it primarily focuses on a prompting method. However, the paper provides extensive experiments demonstrating the effectiveness of the proposed "conversational few-shot prompting method," making it an acceptable approach.
- Regarding the proposed method, the conversational few-shot prompting method appears to address problems by presenting contextually similar topics in a dialogue format. Could you clarify the difference between few-shot prompting in a chat-based model (as shown in the bottom right of Figure 1) and "conversational few-shot prompting"?
- I do not fully understand the motivation for this paper. Specifically, could the authors elaborate on why they believe few-shot prompting is beneficial for instruction-tuned chat models, given that these models are not specifically optimized for this approach?
- In Table 1, a detailed analysis would be beneficial to explain why the performance decreases with larger models as the number of few-shot examples increases. For instance, perhaps larger models exhibit a higher tendency to refuse generating responses. Additionally, in the BoolQ dataset, there is no performance difference in the LLaMA 70B model regardless of the few-shot count, suggesting that the proposed few-shot prompting method may not be effective in this case. This aspect requires further explanation.
- In Table 2, on the MMLU dataset, the LLaMA-3.2-1B model achieves a score of 0, even with varying numbers of few-shot examples. Could you present one or two generated examples? (This part does not impact the overall score).

**Questions:**

Please refer to Weaknesses.

---

### Official Review · Reviewer_cFLZ · 2024-11-04

**Soundness:** 2
**Presentation:** 2
**Contribution:** 3
**Rating:** 5
**Confidence:** 3

**Summary:**

This paper introduces a novel conversational few-shot prompting technique for instruction-tuned chat language models. The proposed approach structures the few-shot examples as a multi-turn dialogue between the user and the assistant, in contrast to the traditional single-prompt format. The authors demonstrate through extensive experiments that this conversational framing significantly improves performance, particularly in low-shot scenarios, across various benchmarks and model families. The results suggest that this method provides a more flexible and robust way to leverage few-shot examples in chat models, enhancing their instruction-following abilities and generalization without the need for additional fine-tuning.

**Strengths:**

(1) The authors propose a conversational few-shot prompting technique that aligns more naturally with the interactive nature of chat models, allowing them to better leverage their inherent conversational capabilities.
(2) The experimental evaluation is comprehensive, covering a diverse set of benchmarks, model families, and shot settings, providing strong empirical evidence for the effectiveness of the proposed approach.
(3) The paper is well-structured, with a clear problem definition, thorough background on few-shot prompting, and a detailed description of the new conversational method, making it easy for readers to follow and understand the contributions.

**Weaknesses:**

(1) The paper lacks detailed implementation details, such as the specific prompting templates used, which may make it difficult for readers to reproduce the experiments.
(2) While the authors demonstrate significant performance improvements, the underlying reasons for the effectiveness of the conversational framing are not fully explored or discussed. A deeper analysis of the model's behavior and the characteristics that contribute to the performance gains would be beneficial.
(3) The paper does not provide any ablation studies or comparisons with other few-shot prompting techniques beyond the traditional single-prompt format, which would help to better understand the specific advantages and limitations of the proposed approach.

**Questions:**

(1) Could you please provide the specific prompting templates used in your experiments, including the format of the few-shot examples and the instructions given to the models?
(2) Can you elaborate on the potential reasons why the conversational framing leads to superior performance, particularly in low-shot scenarios? What are the key characteristics or capabilities of the chat models that enable them to benefit from this approach?
(3) Would it be possible to conduct additional experiments comparing the proposed method with other few-shot prompting techniques, such as prompt tuning or prompted fine-tuning? This could help to better understand the relative merits and limitations of the conversational approach.

---

### Author Response · Authors · 2024-11-25

Dear Reviewers,

We sincerely thank you for your efforts in reviewing our work and valuable feedbacks. We have carefully responded to your comments in our rebuttal and made corresponding revisions to make our paper more rigorous. We would be grateful if you could review our responses. Please let us know if there requires further clarification.

Thank you for your time and consideration.

Authors

---

### Meta-Review · Area_Chair_uz1s · 2024-12-15

**Metareview:**

This paper introduces Conversational Few-Shot Prompting, a new technique that structures few-shot examples as multi-turn dialogues between the user and the AI assistant. By aligning with the interactive nature of chat models, this approach enhances instruction-following capabilities and generalization, particularly in low-shot scenarios, across diverse benchmarks and model families. Extensive experiments demonstrate that this method outperforms traditional single-prompt formats, offering a robust and flexible alternative to improve performance without additional fine-tuning.

Strengths:

This paper introduces a novel conversational few-shot prompting approach that aligns seamlessly with the interactive nature of chat-based LLMs, enabling them to better utilize their conversational capabilities. The experimental evaluation is thorough, spanning various benchmarks, model families, and settings, and provides compelling evidence of the method's effectiveness. Additionally, the paper is well-written, with a clear problem definition, robust background discussion, and detailed methodology.

The reviewers noted various weaknesses, but the two main ones are as follows:

1. Weak analysis: Although the paper shows notable performance improvements, it does not thoroughly investigate or explain why conversational framing is effective.
2. Limited novelty: The paper has limited technical novelty, as it primarily focuses on prompting, and similar methods already exist (e.g., incorporating user and assistant tokens in prompts).

Overall, the combination of weak analysis and limited novelty is, in my opinion, sufficient to recommend rejecting this submission. While the authors' latest response to all reviewers did offer elements that could address Weakness 1, I believe its analysis and discussion of the theoretical foundations should have been a central part of the initial submission, considering this is a submission to an ML conference. As it stands, the paper reads more like an application paper and it was seemingly not updated.

**Additional Comments On Reviewer Discussion:**

All reviewers participated in the discussion, but they all finally sided with rejecting the paper.

One part of this discussion is worth highlighting here, specifically between Reviewer VjQ9 and the authors. In a debate about novelty, the reviewer referred to sources that are not academic publications, which the authors rightly pointed out. While the authors "may be excused for not knowing about papers not published in peer-reviewed conference proceedings or journals" (as per reviewer guidelines), it seems the reviewer's intent was not to suggest that the authors should have cited or compared against specific papers. Instead, the point is that papers are often judged by their level of novelty or creativity, and referencing the existence of similar methods described even in blogs or LLM documentation is a reasonable way to argue that the method proposed in the paper is not particularly innovative. The authors could have compensated for this lack of creativity with a paper that excels in analysis, but unfortunately, that is not the case here.

---

### Decision · Program_Chairs · 2025-01-22

Reject